# Development and Characterization of Natural-Fiber-Based Composite Panels

**DOI:** 10.3390/polym14102079

**Published:** 2022-05-19

**Authors:** Swaroop Narayanan Nair, Aravind Dasari

**Affiliations:** School of Materials Science and Engineering (Blk. N4.1), Nanyang Technological University, 50 Nanyang Avenue, Singapore 639798, Singapore; swaroop.n@ntu.edu.sg

**Keywords:** natural fibers, composite panels, sound absorption, noise reduction, warm water testing

## Abstract

The emphasis on sustainability in materials related to the construction and transportation sectors has renewed interest in the usage of natural fibers. In this manuscript, a different perspective is taken in adopting oil palm fibers (OPF) to develop composite panels and understand their acoustic, mechanical, and water susceptibility (including warm water analysis) properties to provide an insight into the potential of these panels for further exploration. The binder for these composite panels is a water-based acrylic resin, and for reinforcement purposes, fly ash and other metal oxides are used. It is shown that the presence of fibers positively influences the acoustic absorption coefficient in the critical mid-frequency range of 1000–3000 Hz. Even the noise reduction coefficient values highlighting the octave band are higher by more than 50% in the presence of fibers as compared to traditional refractory boards. Quasistatic indentation and drop-weight tests have also highlighted the excellent performance of the composite panels developed in this work. Though the water immersion tests on composite panels and subsequent analysis showed relatively minor changes in their performance, the immersion of the panels in caustic warm water for 56 days has resulted in their severe degradation with a loss of more than 65% in flexural strength.

## 1. Introduction

In the recent past, there has been a growing emphasis on the usage of sustainability concepts in many sectors, including automotive, buildings, and construction. This was largely driven by global ecological and climatic issues as well as resource depletion [1,2]. This has resulted in a renewed interest in the usage of natural fibers and other timber-based systems, which has been backed up by the life cycle analysis. For example, seat backs, interior linings, and panels of many car models from major companies such as Audi, BMW, Mercedes-Benz, Volkswagen, and Toyota have been using natural fiber composites [3]. Indeed, these are applications that do not require the highest mechanical performance, and only a minimum tensile strength of ~25 MPa was specified [4]. Efforts are also underway to replace synthetic fibers such as polypropylene (PP) and polyethylene (PE) with natural fibers such as hemp, palm, and jute in ultra-high-performance concrete (UHPC) [5,6]. As UHPC, due to its dense microstructure and high compressive strength, is vulnerable to thermal spalling, the addition of PP or PE fibers into concrete is a common practice to prevent explosive spalling during a fire scenario. Apart from these, partition/lining boards and thermal barrier panels are other applications with potential for these natural fiber-based composite systems.

However, it is important to acknowledge the drawbacks of using natural fibers in applications that are exposed to external weathering conditions or that demand high consistency in the performance. As well known, swelling of fibers is a major consequence of their susceptibility to moisture/water because of the presence of excessive hydroxyl groups (for example, see [7,8,9,10,11]). Despite many efforts in treating fibers (silane, acetylation, or even usage of maleic anhydride-based polymers) before incorporating them into different matrices (even non-polar matrices such as PP), the commercial success still awaits. Even in our previous investigation on the usage of natural fibers in UHPC, we have observed that accelerated weathering testing of these samples for 30 days has reduced the compressive strength by up to 10% compared to unweathered samples with a fiber loading of 10 kg/m^3^ [12]. Accelerated weathering test conditions include exposure of the samples to 60 cycles, with each cycle representing an 8 h of UV exposure at 60 °C (irradiance of 1.55 W/m^2^), a 0.25 h water spray, and 3.75 h of condensation. Similarly, Shinoj et al. [13], in their study on low-density polyethylene (LDPE)/OPF-based composites, have observed an increase in the thickness (swelling) of the fibers by up to 15% after the water immersion process. This is irrespective of the various fiber sizes used and their loading levels. In another similar study, Le Duigou et al. [14] conducted an 18-month long seawater exposure study on flax/PLA bio-composites and reported ~10% and ~20% reduction in tensile stiffness and flexural strength, respectively. In short, fiber swelling has been reported as one of the major reasons for the reduction in mechanical properties of natural fiber composites after long-term environmental exposure. The swollen fibers are expected to induce cracks and overstress the surrounding matrix regions [15].

In this study, an exploratory investigation is carried out on the development and characterization of OPF-based composite panels. The fiber mats were pretreated before incorporating with a water-based acrylic resin. A range of different mechanical and acoustic properties were studied to establish the potential of these materials in the construction and transportation sectors.

## 2. Experimental Work

### 2.1. Materials

Palm fibers in the form of woven mats with a density of 2000 gsm (~18 mm thick) were obtained from a Malaysian plantation through a Singapore distributor, Evergreen Consultancy and Resources Pte Ltd. A water-based acrylic resin Plextol R4180 was purchased from Synthomer Sdn Bhd., Johor, Malaysia, and has total solid content of 49–51%. Sodium hydroxide pellets reagent grade, ≥98% purity (NaOH), was obtained from Sigma-Aldrich, Singapore. In order to make the composite panels more sustainable, F-grade fly ash (FA) with average particle size ~40 μm and density 2.38 g/cm^3^ was used as a reinforcing filler in one of the formulations. FA was purchased from Jaycee, India. Other powder additives, including polyphosphates (used to modify the viscosity of the resin during the casting process) and aluminum-based oxides (for improving the mechanical properties of the resultant boards), were also used.

### 2.2. Surface Modifications of Fibers

As mercerization or alkali treatment of natural fibers has been recognized as an efficient methodology for cleaning and removal of any residual oil contents (for instance, [16]), it has been adopted here. Mercerization helps to improve the fiber matrix physiochemical interaction through interface fiber surface modification [17]. A simple reaction scheme for this process is shown in Equation (1). It is known that mercerization of natural fibers will increase the amount of cellulose II (anti-parallel chain conformation) at the expense of cellulose I (parallel chain arrangement), which is an irreversible polymorphic transformation. This will also be associated with a drop in percentage crystallinity of the fibers, removal of hemi-cellulose, impurities, and oily residues, as well as roughening of the fiber surfaces.
Fiber − OH + NaOH → Fiber − O^−^ − Na^+^ + H_2_O(1)

The commercially obtained fiber mats were washed with tap water before completely immersing them in 2.5 wt.% of NaOH solution for 24 h at room temperature. Subsequently, they were washed with running tap water to remove any residual alkali deposits. The washed mats were then dried in an oven at 60 °C for up to 24 h. Weight reduction of up to 22–25% was observed after the mercerization process compared to the original fiber mat (also due to the removal of loose fibers from mat). It is important to note that the optimum 2.5 wt.% concentration of alkali was chosen after several iterations and experiments. An increased percentage of alkali after an optimum level weakens the fiber strength. For example, when date palm fibers were treated with 10 wt.% of alkali solution, severe degradation of the fiber tensile strength (up to 45% reduction) was observed [18].

### 2.3. Casting of Composite Panels

A rectangular stainless-steel mold of dimensions 420 mm × 310 mm × 20 mm was used for the casting of composite panels. The fiber mats were cut to exact shape of the mold, and weight of the mats was noted. Two different formulations (with and without FA) were designed and adopted in this study. In the formulation without FA, solid acrylic resin content proportion was maintained at 20 wt.% with the rest including OPF fiber, ~25 wt.% of metal oxide, and minor inclusion of phosphates. In the other formulation, the only change is the incorporation of ~25 wt.% of FA substituting the metal-based oxides. For the preparation of the panels, to ensure a homogeneous distribution of fly ash, premixing of the powder fillers was carried out before mixing with the resin. Subsequently, the weighted amounts of the resin and additives were mixed using a mechanical spindle mixer. The mixture was poured over the pretreated fiber mat placed within the mold, and a trowel-and-tap method was used to make sure the water-based resin wets the fiber mat thoroughly. Top surface of the mold was covered with porous Teflon ply layer that allows the moisture/vapor to escape during curing. Curing was carried out in a convection oven at 60 °C for 4 h, followed by 80 °C for another 4 h. An additional post-curing at 100 °C for an hour was then carried out to remove any left-over moisture. It should be noted that these curing conditions were selected after several trials to eliminate cracking, blisters, and high degree of porosity. After curing, the panels were allowed to cool down to room temperature before demolding. 

### 2.4. Surface Characterization of Natural Fibers

Fourier transform infrared spectroscopy (FTIR) of both treated and untreated fibers was carried out in the range of 4000 to 400 cm^−1^ using Perkin Elmer Spectrometer, in the attenuated total reflectance (ATR) mode. A total of 32 scans were attained each time at a resolution of 4 cm^−1^. Scanning electron microscopy was also conducted on the treated and untreated fibers using the JEOL JSM-5500LV. The spot size was set at 6 with accelerating voltage ranging from 5–20 kV and a physical working distance of 15 mm.

### 2.5. Moisture Movement, Water Absorption, and Warm Water Testing

All these tests (moisture movement, water absorption, and warm water testing of fibers/mats and composite panels) were performed as per ASTM C1185-08 [19] standard. For the moisture movement analysis, both individual fibers (~5 cm in length) and mats (dimensions of 5 cm × 5 cm) were analyzed. Before the test, the specimens were dried in a convection oven at 80 °C for 30 min and subsequently conditioned the specimen to a practical equilibrium at ~23 °C, 30% relative humidity (RH) for 24 h followed by ~23 °C, 90% RH for another 24 h period. The change in moisture content is measured as a percentage change in weight with respect to RH.

Similarly, for water absorption analysis, dried fiber samples and composite panels were immersed in a water bath at room temperature. The weight of samples was monitored at specific time intervals until equilibrium was reached. The percentage change in weight was reported. The test will help to understand the water retention capacity of fibers and the composite panel.

Compared to water absorption test, warm water test is more rigorous and time-consuming. This test helps to evaluate the mechanical performance of the composite panel if it were to be used under wet conditions. Warm water test was carried out on composite panel of dimensions 300 mm × 130 mm × 20 mm. Prior to the test, the specimens were dried in an oven for 6 h at 80 °C before recording their weight. Subsequently, the specimens were completely immersed in water bath (containing excess aq. Ca(OH)_2_) maintained at 60 ± 2 °C for 56 ± 2 days. Thereafter, specimens were conditioned at 23 ± 2 °C and relative humidity of 50% ± 5% for 2 days before measuring their final weight. Subsequently, flexural analysis was carried out on the specimens to evaluate the changes in mechanical properties.

In order to study the absorption rate under warm water conditions, diffusion rate was calculated according to Fick’s law. Very similar to water absorption test, percentage weight gain ($W_{t}$) was calculated over the test period at regular intervals and plotted against square root of water immersion time [20,21] based on Equation (2).
(2)$$S_{C}=\frac{W_{t}}{W_{\propto }}=4\sqrt{\frac{Dt}{\pi h^{2}}}$$
where $S_{C}$ is the sorption coefficient, *D* is the diffusion coefficient, $W_{t}$ represents the percentage weight gain of the sample at each absorption stage, *t* is the time, and $W_{\propto }$ represents the saturated percentage weight gain. In order to identify the mechanism behind the variations in absorption rate (Fickian diffusion, relaxation controlled, and non-Fickian or anomalous), theoretical distinction is made based on the slope of Equation (3) [22,23]. It should be noted that the absorption mechanism depends upon the polymer structure, nature of the fiber, type of fillers, and interfacial adhesion, apart from defects introduced during the manufacturing process.
(3)$$log\left(\frac{W_{t}}{W_{\propto }}\right)=log\left(k\right)+nlog\left(t\right)$$
where k and n are constants. For Fickian diffusion n=0.5, in relaxation n<0.5, and for anomalous mechanism 0.5<n<1. The diffusion coefficient is calculated using Equation (4).
(4)$$D=\pi {\left(\frac{kh}{4W_{\propto }}\right)}^{2}$$
where *h* is the thickness of the sample and *k* is the slope of the plot between $W_{t}$ and $\sqrt{t}$.

### 2.6. Mechanical Testing of Composite Panels

#### 2.6.1. Three-Point Bending

Flexural strength of the composite samples was determined as per ASTM C1185-08 [19] standard in three-point bending mode. The specimens of dimensions 300 mm × 150 mm × 20 mm were cut from the casted panels and conditioned at 23 ± 1 °C and 50% RH for a week before testing on an Instron 5567 machine. A load cell of 30 kN was used and calibrated before testing. A fixed span length of 254 mm was maintained for testing.

#### 2.6.2. Quasistatic Indentation Testing

In order to understand the resistance to penetration and energy absorption of the composite panels, quasistatic indentation tests were carried out on panels with dimensions of 100 mm × 100 mm. The quasistatic tests were conducted on Instron 5567 with an indenter setup. The test jig consists of a base steel structure and a steel clamping plate at the top with a hole of 76.4 mm diameter at the center of the clamp. Test specimen was mounted in between those steel clamps. A hemispherical steel tup of 13 mm diameter was used as the indenter. The loading was controlled with a constant head speed of 0.5 mm/min.

#### 2.6.3. Impact Testing

Drop weight impact tests were also conducted on the composites to further understand their energy absorption behavior during an impact. The test was conducted on Cadex Twin wire 1000 kg machine, Quebec, Canada with a spherical impactor (radius 73 mm) made of plated steel. The impactor (projectile and the attached carriage) was connected to an accelerometer, and its velocity before the impact was measured using a photodiode placed above the sample. Specimens of dimensions 100 mm × 150 mm were clamped to the bottom fixture, which was connected to the load sensor. Prior to the test, grid lines were drawn on the specimens’ top surfaces with a marker to understand ‘visually’ the extent of damage after the test.

### 2.7. Acoustic Absorption Properties

Sound absorption coefficient of the composite panels was evaluated in accordance with ASTM E-1050 [24] standard using Type 4206 impedance tube kit from Bruel and Kjaer, Virum, Denmark. The equipment consists of two microphones, an impedance tube, and an amplifier. Sound waves were produced from a source inside the impedance tube and microphones mounted inside the tube measured the sound pressure level on both sides of the specimen. Two different sizes of impedance tubes were used in this work to cover a frequency range of 50–6400 Hz. The large tube with 100 mm diameter measures the sound absorption coefficient at a lower frequency range from 50–1500 Hz, while the small tube with 29 mm diameter measures from 500–6400 Hz. Samples were cut to fit exactly the 100 mm and 29 mm tubes. Three samples from each formulation were tested in the impedance tube.

### 2.8. Dynamic Mechanical Analysis (DMA)

Parameters such as storage modulus, loss modulus, and mechanical damping factor as a function of temperature and frequency of the panels were measured using TA Instruments DMA Q800 equipment. Thin panels of dimensions 60 mm × 14 mm × 7 mm were prepared and cured in a similar manner as described earlier for this purpose. The test was conducted in three-point bending mode. The amount of energy absorbed and dissipated was calculated using the measured strain. The material response was monitored over a range of frequencies at a constant amplitude of deformation. A step-by-step increase in the temperature from 35 °C to 150 °C with an increment of 5 °C and a frequency sweep from 1–20 Hz establishing 8 points per decade at a constant amplitude of 15 μm were chosen as input parameters. 

## 3. Results and Discussion

### 3.1. Mercerization of Fibers

FTIR spectra of the fibers before and after the alkali treatment are shown in Figure 1a. Irrespective of the treatment, a broad peak between 3600 cm^−1^ to 3000 cm^−1^ attributable to O-H stretching; and peaks around 2970 cm^−1^ to 2860 cm^−1^ corresponding to aliphatic C-H stretching are evident. The major differences between the two spectra are in the region between 1800 cm^−1^ to 1100 cm^−1^. The absence of the sharp and intense peak with a maximum of around 1730 cm^−1^ (corresponding to a carbonyl group (-C=O)) in the treated fibers suggests successful removal of hemicellulose. This is also confirmed by the absence of a C-O stretching peak in the O=C-O group of hemicellulose at 1231 cm^−1^. Further, as expected, after the alkali treatment, the surface of fibers became rougher due to the dissolution and removal of hemicellulose; pectin; and other impurities (Figure 1b,c). This increases the interfibrillar region as compared to untreated fiber that shows a smooth surface. This rough surface increases the surface area and is expected to benefit interfacial adhesion with the resin.

### 3.2. Analysis of Acoustic and Damping Properties of the Panels

The results of a representative set of samples obtained from the impedance tube setup experiments for both long and short tubes are shown in Figure 2 by eliminating the noise signal using the Fast Fourier Transform (FFT) function. For comparison purposes, commercially bought gypsum (Gyproc with a density of ~900 kg/m^3^ and thickness of 19 mm) and magnesia (from Davco, with a density of ~1025 kg/m^3^ and thickness of 19 mm) boards were also tested under similar conditions as the composite panels (densities of ~1450 kg/m^3^ and ~1100 kg/m^3^ and thicknesses 20 mm each with and without FA panels, respectively). Three samples from each formulation were tested in the impedance tube. It should be noted that the sound absorption coefficient (SAC, α) points to the magnitude of sound absorbed and is defined as the ratio of absorbed energy (*E_a_*) to the energy of the incident wave (*E_i_*). That is, α = $\frac{E_{a}}{E_{i}}$.

As opposed to the active noise canceling/controlling concept that works on the principle of generating a counter wave and its superimposition on the target signal [25,26], passive noise control systems such as the composite panels reduce the noise by directly interacting with the acoustic waves through either ‘insulation’ or ‘absorption’. ‘Insulation’ generally works on the principle of mass law; that is, the higher the mass, the higher the insulation ability. In terms of acoustic absorption, in addition to the thickness, viscoelastic properties of materials do play a significant role. However, these mechanisms are particularly effective at higher frequencies than at lower frequencies. Their performance at lower frequencies is generally related to thickness and interconnectivity in the pores [27]. Based on the literature, it is noted that passive noise controlling systems generally demonstrate good performance at high frequencies (>1000 Hz), while active noise cancellation is relatively more effective at lower frequency regions of up to 500 Hz [26,27]. Indeed, as evident from Figure 2 in the presence of fibers, the critical mid-frequency range of 1000–3000 Hz is influenced by an α value of ~0.65. In this frequency range, there are no absorption peaks for gypsum and magnesia boards. They exhibit only a strong peak with an α value of ~0.7 at ~4000 Hz. Apart from this, considering the thickness of the panels employed, all of them do show absorption peaks even at lower frequencies of 200–400 Hz. However, with natural fibers and without FA, the α value is 0.98 at 300 Hz. Gypsum and magnesia panels show peaks with α values of 0.7 and 0.58, respectively, at the frequency of 300 Hz.

Another way to characterize the sound absorption efficiency of these panels is to calculate the Noise Reduction Coefficient (NRC) [28], which is the arithmetic average of the absorption coefficient values measured at the octave band of 250 Hz, 500 Hz, 1000 Hz, and 2000 Hz. As shown in Table 1, the presence of natural fibers shows better NRC values than magnesia and gypsum panels.

As discussed earlier, based on the literature on fiber reinforced composites, it was noted that thickness of the panel, viscoelastic properties of the system, fibrous microstructure, and multi-scale micro-morphology are critical in governing the damping ability or sound energy dissipation of the panels [28,29,30,31]. The chemical nature of the matrix is another critical parameter that influences sound absorption. Yang and Li [29] have shown that irrespective of the type of fiber (jute, ramie, carbon, glass, and flax) and with epoxy as the binder, there was a broad peak in the range of 2000–8000 Hz with a peak SAC value of ~0.35–0.40 around 5800 Hz. However, when the fibers were tested by themselves, it was found that within the frequency range of 250–2000 Hz, jute, ramie, carbon, glass, and flax fibers showed different SAC values of 0.65, 0.60, 0.45, 0.35, and 0.65, respectively. These differences again illustrate the importance and dominance of binder nature (stiffness, the bulkiness of the groups, interaction with the fibers, etc.) in influencing (or compromising) the acoustic properties of the composites. Though in relation to the current study, acrylic has a relatively lower *Tg* (~33 °C), considering its content of 20 wt.%, the presence of inorganic fillers will have a determining influence on the sound absorption behavior of these panels.

For polymer/fiber composites, when a sound wave enters the system, the pressurized air molecules move through the viscous matrix/pores, and the elastic stress wave is carried through the solid backbone of the polymer material [32]. As a result of this, internal friction builds up at the interface of the air and the fiber surface (in fiber reinforced composites) or the air and the cell wall (in polymer foams). This is dissipated as heat loss. That is, the mechanisms are indicated as coulomb (inadequate interface friction between matrix and fiber), viscous (dissipation of heat because of the flow of the medium), and hysteric (buckling, distortion, etc. of the solid resulting in heat loss) damping [33]. Even for polymer nanocomposite foams, it was hypothesized that the enhancement in sound absorption might be due to increased sound scattering and greater heat dissipation at the polymer-filler interface [30].

Considering the above discussions, it is not surprising that the panel without the reinforcing fly ash showed the best sound absorption even at lower frequencies as compared to other stiff panels/boards. The alkaline treatment of the fibers is also expected to improve the dissipation extent due to the rougher surfaces. Figure 3 shows the storage modulus, loss modulus, and damping ratio (*tan δ*, ratio of loss modulus to storage modulus) plots for both the composite panels. At tested frequencies of 1 Hz and 20 Hz, the panel without FA shows a relatively higher damping ratio as compared to the panel with FA until temperatures of up to 70 °C. At 20 Hz, the temperature corresponding to the peak of tan δ is ~45 °C for the formulation without FA. With FA, within the temperature range tested, the peak is not clearly identified (on the rise even at 30 °C). This is true for both systems when tested at a frequency of 1 Hz. When tested at lower frequencies, the density (and mass) should play an important role. However, at higher frequencies, the material and morphological characteristics of the composite panel will also influence the dissipation process. Beyond 70 °C, irrespective of the frequency, the differences in damping are minimal and correlate well with the corresponding changes in the loss modulus as the molecular motion within the material increases. Regarding storage modulus, on the other hand, though reduced with temperature as expected, the drop is not significant. Furthermore, not surprisingly, the panel with FA shows higher *E*′ values at all the measured temperatures. It is also evident that *E*′ is higher at 20 Hz as compared to 1 Hz. This behavior is noticed with many fiber-reinforced composite systems. At higher frequencies, the relentless oscillation causes the molecular chain to attain a shorter time to relax after each oscillation and makes them highly elastic, affecting the *E*′ value [31].

### 3.3. Mechanical Properties of Panels and Susceptibility to Moisture/Water

#### 3.3.1. Moisture and Water Absorption Test

As discussed earlier, the susceptibility of natural fiber-based composites to moisture/water has been well reported in the literature for different matrices [7,8,9,10,11,34,35]. Recently, Sanjeevi et al. [35] have reported that for a hybrid phenol formaldehyde system with Areca fine fibers and Calotropis Gigantea fibers, the higher the natural fiber content in the composite, the higher was the water absorption rate. They have noted a 4.3% water absorption with 45 wt.% of fiber loading. It has dropped to 3.6% with 25 wt.% loading of the fibers. A few other studies have taken advantage of this behavior by correlating it with the dielectric properties of the system [36,37]. For instance, Fraga et al. [37] have tested the changes in the dielectric permittivity with water absorption in the frequency range of 200 Hz to 1 MHz. The dielectric constant of the composites has decreased with frequency and increased with water uptake. Nonetheless, it is agreed that water uptake will generally degrade the system through interfacial cracking, ultimately leading to a significant reduction in mechanical properties [38]. Therefore, the repeated swelling and deswelling (when exposed to moisture/water and subsequent drying), in fact, could defeat the purpose of improving the interfacial interaction of the fibers with the matrix. Table 2 shows the susceptibility of the samples to moisture uptake and water absorption/retention. Moisture uptake for individual fibers and mat is not significantly different. Furthermore, no major differences are seen between the alkaline treated and untreated fibers/mats. However, when completely immersed in water until an equilibrium is reached in terms of water uptake/retention, a weight gain of 25.2% is noted for alkaline treated mats and 22.7% gain in the case of untreated mats. More importantly, with the composite panels, a weight gain of 24.7% and 30% is noted for panels without and with FA, respectively. The pozzolanic nature of FA could be a source of difference. As evident, weight gain without FA is consistent with what is expected based on the water absorption capacity of fiber mats by themselves. Though the composite panels are generally expected to have better resistance to water absorption, the resin percentage and the chemical nature of the resin are critical for that. In the current work, the panels consist of only 20 wt.% of resin, and therefore, it is understandable that there will be easier pathways for water to access the fibers. Moreover, the poor interface between fibers and matrix is a great source of water retention. Despite the higher content of water absorbed, more importantly, there were no visual signs of any damage on the panels, irrespective of the presence or absence of fly ash.

#### 3.3.2. Warm Water Testing

Understandably, higher temperatures will accelerate the water uptake and the rate of approach towards saturation (equilibrium) [8,11]. The diffusion coefficient of water, pulling of water molecules into the system because of surface groups of natural fibers, and *T_g_* of the resin are factors that contribute to this, apart from interfacial cracks/porosity generated because of swelling of fibers. In the current work, warm water tests were conducted for the composite panels for 56 days at 60 °C in a caustic environment (with excess lime added to the water). Over the 56 days of the test period, the panel without FA shows a weight gain of 35.7 wt.%, and with FA, it is 43.2 wt.%. Sediments were found inside the test tray, which shows the signs of degradation of the material. The specimens have noticeable cracks over the entire surface and appear to be flexible when taken out of the lime water after 56 days of testing. After the warm water test, the specimens were left to dry for at least a couple of days before subjecting them to flexural tests.

Figure 4 shows the ratio of %weight gain at each measurement during the 56 days to the saturated %weight gain plotted against the square root of time (s). From the diffusion data analysis, the slope of the water uptake profile can be split into two regions (termed here as primary and secondary absorption stages). There seems to be a significant increase in water uptake within the first 7 days. A ratio of ~0.7 is noted within 4 days of the test and changed to ~0.8 in 7 days after the start of the test. Corresponding %weight gain (*W_∞_*_1_) is listed in Table 3. However, after this, there was a significant reduction in the uptake, and it took almost 51 days to change the ratio from 0.8–1.0. Based on the n value, it is found that the primary stage behavior is anomalous (until the first 7 days), and thereafter, the rest can be modeled as a Fickian diffusion process (Table 3).

#### 3.3.3. Flexural Testing

After the warm water testing, conditioning of the panels was carried out at 23 ± 2 °C and relative humidity of 50% ± 5% for 2 days. Subsequently, a drop in weight by 8–10% was detected. Table 3 also shows the flexural testing data of the composite panels before and after warm water immersion tests. The data clearly illustrate the degradation experienced by the panels (irrespective of the composition) after the warm water tests. This is also evident from the representative force–displacement curves shown in Figure 5. The cracks that have formed during the warm water test have propagated along the tension side of the specimen under flexural loading conditions deteriorating the load carrying capacity of the composites.

#### 3.3.4. Quasistatic Indentation

The force–displacement data of the composite panels during the quasistatic indentation tests are shown in Figure 6. The damage tolerance of the panels is different, and as expected, with FA, it performed better. A peak load of 1460 N (corresponding displacement of 16.4 mm) is noted for the composite panel with FA as compared to ~1215 N (corresponding displacement of ~26 mm) for the panel without FA. The energy absorption calculated by measuring the area under the curves has resulted in average values of ~28.3 J and ~32.4 J for panels without FA and with FA, respectively. During the tests, the impactor penetrated the panels and induced a significant amount of damage on the underside due to fiber breakage, pull-out (bottom), and matrix cracking (Figure 7). Origin of the cracks can be seen on the highly stress concentrated area on the upper surface for both types of panels, but the intensity of cracking is relatively higher in panels without FA.

#### 3.3.5. Drop Weight Test

The calculated energy ‘*E*’ from the quasistatic indentation test was used to determine the height ‘*H*’ of impact (*E* = *mgH*) in these drop weight tests on the composite panels. A hemispherical projectile with a mass ‘*m*’ of 5 kg was chosen, as mentioned earlier. The projectiles were dropped from heights *H*, *H*/2, and *H*/4. The damage experienced by the panels (underside) when the impactor was dropped from the maximum calculated height is shown in Figure 8. When the projectile was dropped from a height of 58 cm, a dent appeared on the upper surface while cracks originated on the underside. A crack was propagated from the point of impact towards the outer direction. When the height of the impactor decreased to 29 cm (*H*/2) and 14.5 cm (*H*/4), no visible damage was seen on the top surface, even in the presence of a few pre-existing hairline cracks after the drop weight test. With FA, the composite panels showed a similar trend, but the damage was more visible with a relatively higher intensity of cracking (at *H* = 66 cm). Similar to the panels without FA, even here, height *H*/2 and *H*/4 have little influence on the FA panels.

## 4. Conclusions

In this exploratory work, OPF-based acrylic composite panels were developed, and their mechanical, water absorption, and acoustic (absorption/damping) properties were characterized. This was necessary to understand their potential in construction and other sectors.

Mercerization process of the fibers and their mats was carried out with 2.5 wt.% of NaOH, which has resulted in the removal of hemicellulose and other impurities along with improving the roughness of the surface of fibers.The presence of natural fibers had a positive effect on the acoustic absorption properties, particularly in the range of 1000–3000 Hz. In this region, no absorption was detected for conventional stiff panels based on gypsum and magnesia. However, they exhibit absorption in the range of 3000–6000 Hz. The noise reduction coefficient also confirmed the importance of the presence of natural fibers in the panels.The composite panels without the reinforcing fly ash have also shown better damping properties than the panel with fly ash until temperatures of 70 °C. With an increase in frequency from 1–20 Hz, a shift of damping peak towards relatively higher temperatures was noted.As the resin content in the composite panels was fixed at 20 wt.%, water uptake of the panels were relatively higher than expected (panel with FA showed 30 wt.%, while the panel without FA showed 24.5 wt.%). Warm water testing has provided clear evidence of the disadvantages of these panels. A huge reduction in flexural strength was noted in the panels (irrespective of the presence or absence of FA) after the warm water testing. A reduction of 75% was observed for the panel with FA, while the panel without FA showed a drop of 69.5%.Quasistatic indentation tests and drop weight impact tests have also shown that the composite panel with FA was stiffer than the panel without FA.

In short, the developed natural-fiber-based composite panels have shown potential for further exploration in construction and automobile sectors with good mechanical and acoustic properties. However, the panels have clearly degraded after warm water testing for 56 days, reaffirming the drawback of the presence of natural fibers in such a high volume when used in wet conditions.

## Figures and Tables

**Figure 1 polymers-14-02079-f001:**
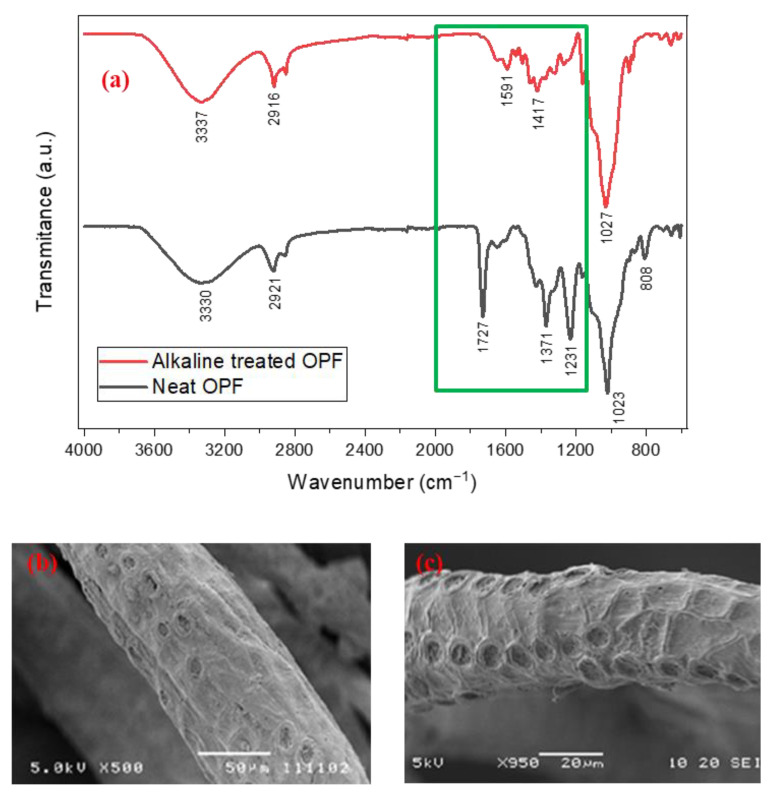
(**a**) FTIR spectra of OPF (untreated and alkaline treated fibers) highlighting the major differences in the region between 1800 cm^−1^ to 1100 cm^−1^; and (**b**,**c**) SEM micrographs showing the surface morphological features of randomly selected (**b**) untreated fiber and (**c**) an alkaline-treated fiber from the mat.

**Figure 2 polymers-14-02079-f002:**
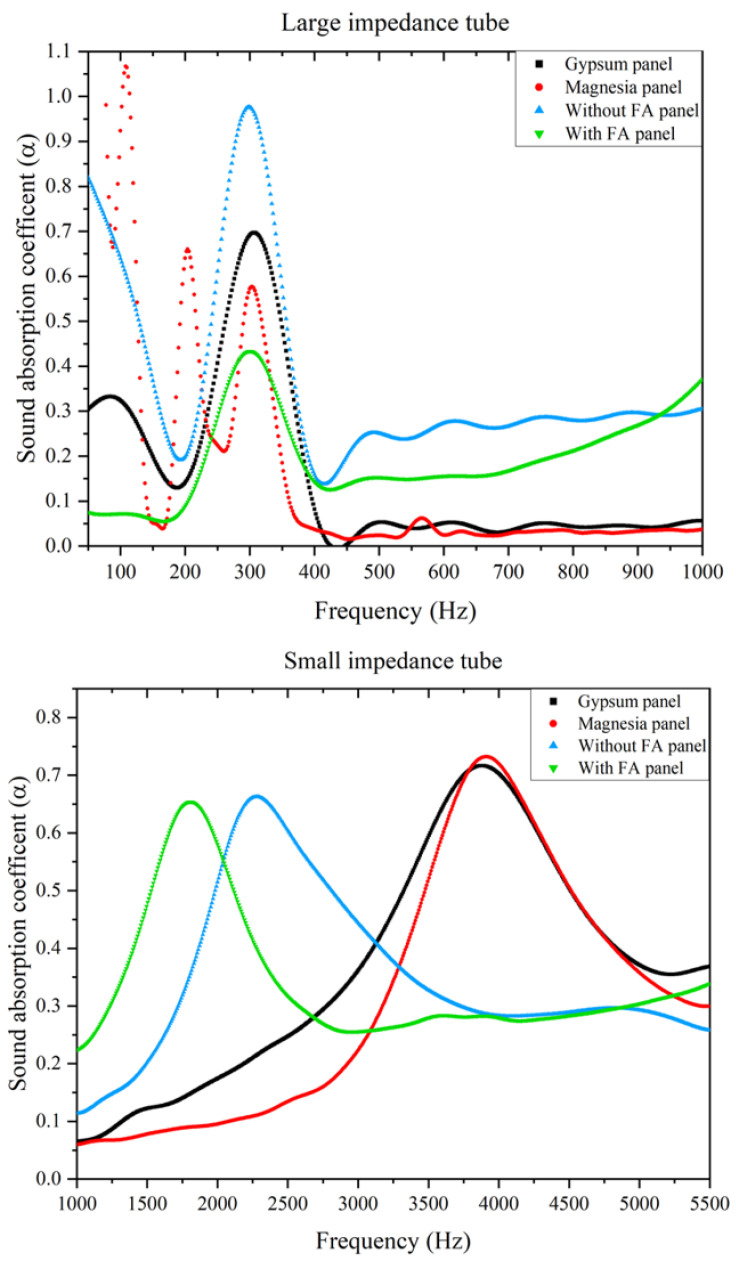
Comparison of the representative data obtained from large and small impedance tubes across a frequency range of 50–6400 Hz for different panels.

**Figure 3 polymers-14-02079-f003:**
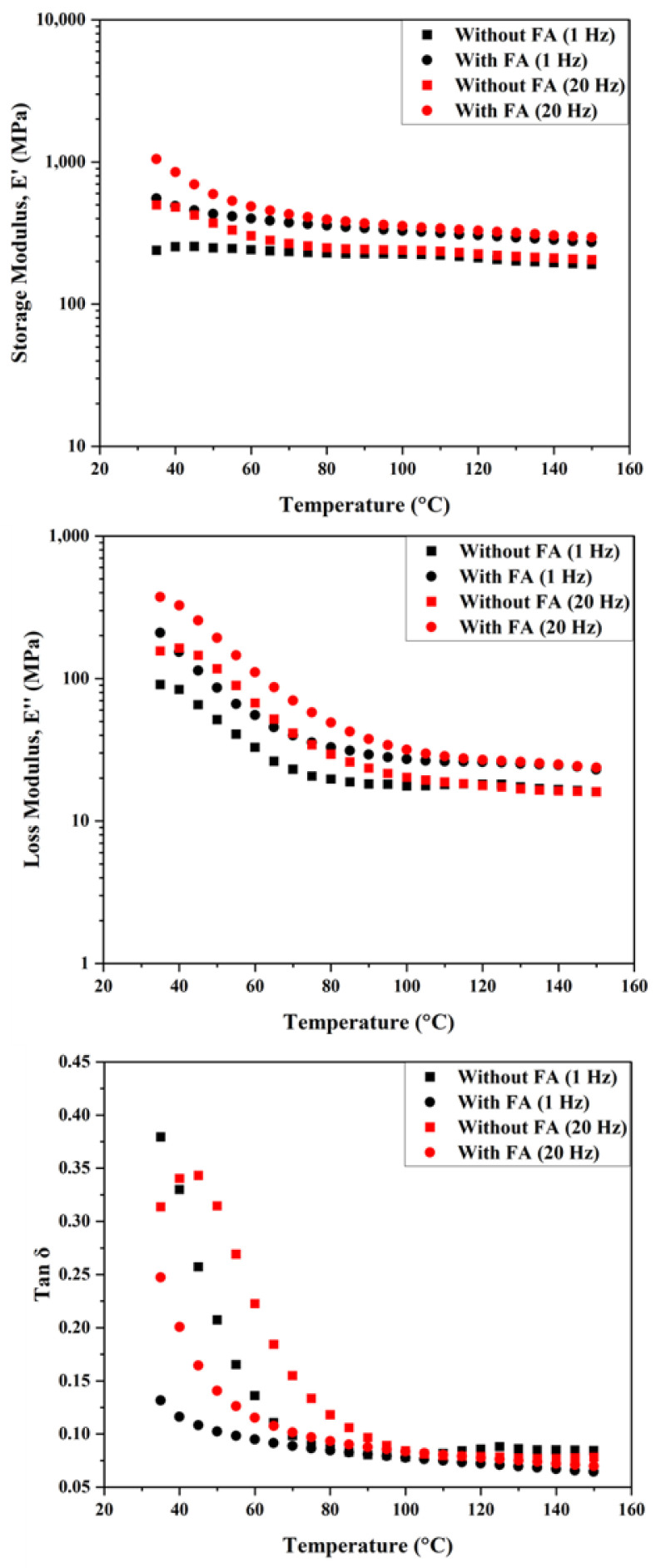
Storage modulus, loss modulus and *tan δ* curves of the composite panels at 1 Hz and 20 Hz.

**Figure 4 polymers-14-02079-f004:**
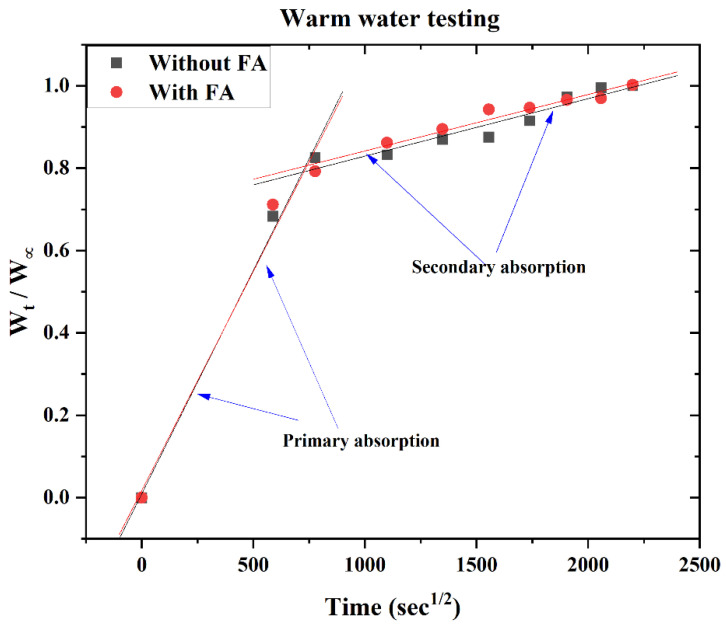
Water uptake behavior in the panels during their warm water testing.

**Figure 5 polymers-14-02079-f005:**
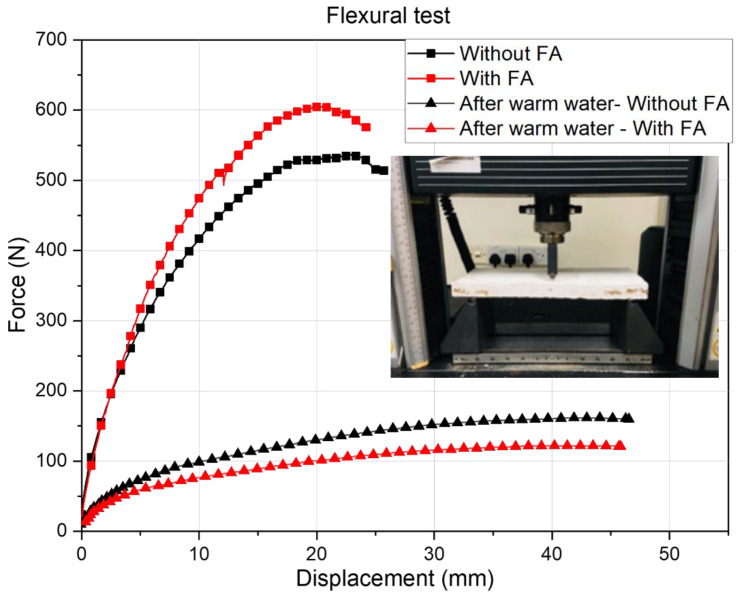
Force–displacement curves of the composite panels before and after warm water immersion test.

**Figure 6 polymers-14-02079-f006:**
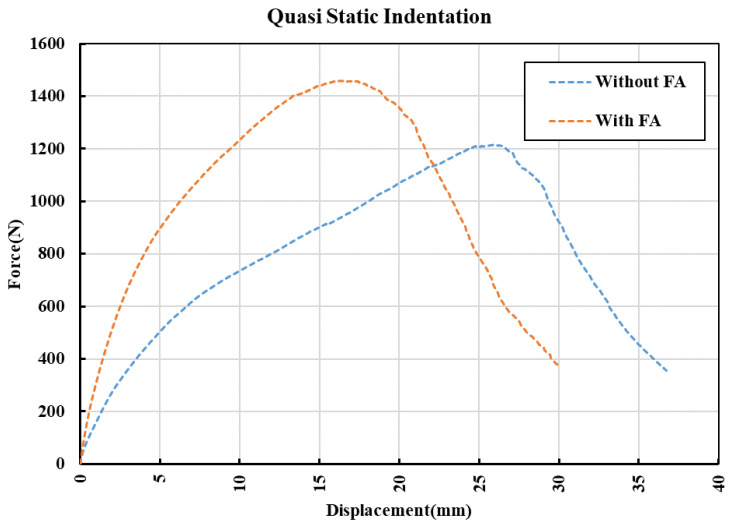
Force versus displacement curves of the composite panels during the quasistatic indentation test.

**Figure 7 polymers-14-02079-f007:**
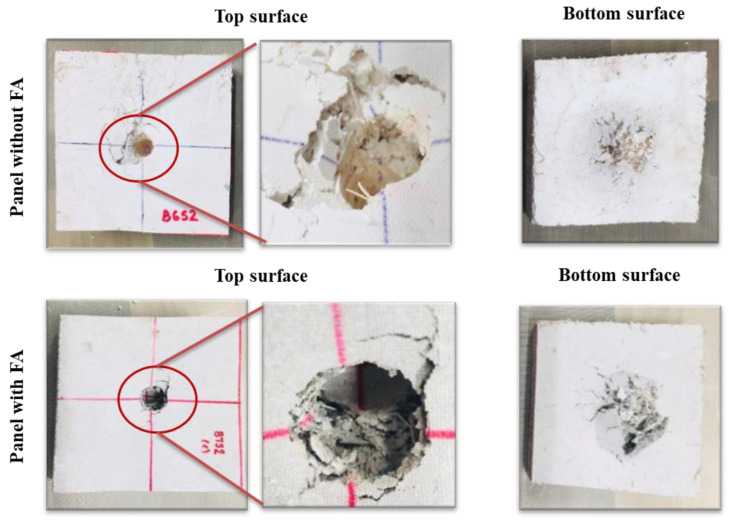
Damage on the composite panels seen after the quasistatic indentation test. The specimen dimensions are 100 mm × 100 mm × 20 mm.

**Figure 8 polymers-14-02079-f008:**
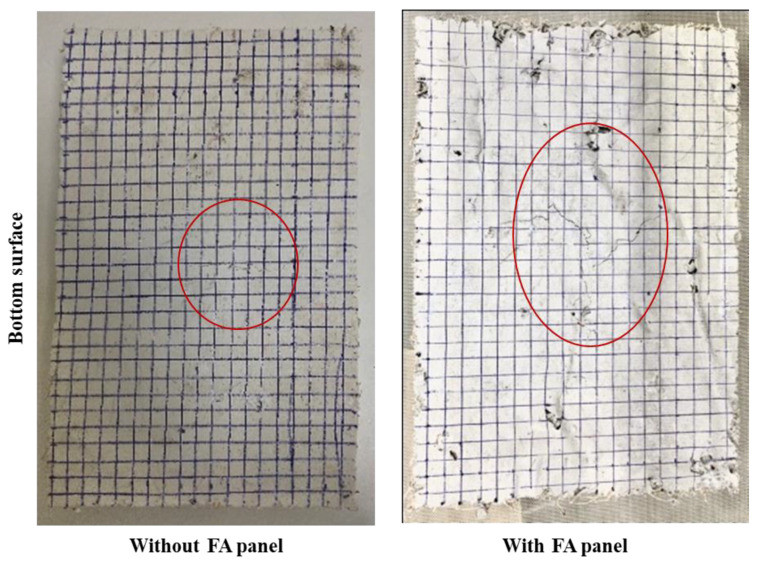
Bottom surfaces of the composite panels after drop weight impact tests (specimen dimensions are 100 mm × 150 mm × 20 mm, and each square represents 5 mm × 5 mm).

**Table 1 polymers-14-02079-t001:** Sound absorption and noise reduction coefficient values of different panels measured at the frequencies of interest (octave band). The average thickness of the panels is also listed.

Material	Average Thickness (mm)	Frequency (Hz)				NRC
		250	500	1000	2000	
Gypsum	19	0.41	0.05	0.07	0.17	0.17
Magnesia	19	0.38	0.03	0.06	0.10	0.14
Without FA	20	0.61	0.25	0.11	0.51	0.37
With FA	20	0.29	0.15	0.22	0.58	0.31

**Table 2 polymers-14-02079-t002:** Water absorption and moisture analysis of the fibers, mats and panels.

	Moisture Uptake (%)		Water Absorption (%)			
	Fibers (Individual)	Mat	Fibers	Mat	Composite	
					Without FA	With FA
Untreated	6.9	6.4	31.6	22.7	−	−
Alkaline treated	6.8	6.3	30.8	25.2	24.5	30

**Table 3 polymers-14-02079-t003:** Warm water testing and flexural analysis.

Composite Panels	Primary Absorption Stage		Secondary Absorption Stage	
	*W_∞_*_1_, %	*D* (×10^−13^ m^2^s^−1^)	*W_∞_*_2_, %	*D* (×10^−13^ m^2^s^−1^)
Without FA	29.5	1.07	4.8	0.65
With FA	34.2	0.75	6.7	0.33
	**Flexural Strength * (MPa)**			
	**Before warm water test**		**After warm water test**	
Without FA	3.21 ± 0.23		0.98 ± 0.08	
With FA	3.35 ± 0.33		0.83 ± 0.17	

* Based on the average of three specimens.

## Data Availability

Data is available from the authors on request.

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
