# Peer review of "Development and Characterization of Natural-Fiber-Based Composite Panels"

_polymers, 2022, doi:10.3390/polym14102079_

Round 1
Reviewer 1 Report
In this manuscript, the authors designed and prepared the oil palm fibers (OPF) filled water-based acrylic composite panels. The research on the acoustic, mechanical and water susceptibility properties of this natural fibers based composite panels is very meaningful. However, the influences of the surface modifications of fibers and fly ash on the above properties are not clearly distinguished and discussed. Moreover, the acrylic composite panels with or without OPF need to be studied to support your opinions. Some other comments are listed as follows:
1: In the fiber surface modifications process, what is the contents of natural fibers and the NaOH solution.
2: In the acoustic absorption measurement (page 5), the sample thickness of disc samples needs to be emphasized. The acoustic absorption properties are highly dependence on sample thickness.
3: In figure 2.7, why the diameter of typical untreated fiber is about 100 μm, but the treated OPF is about only 50 μm. The alkaline treated process seems very intense.
4: In figure 4, only composite panels without FA which tested in 20 Hz exhibited a glass transition at about 50 ℃. Why other three samples do not show Tg peak at the DMA measurements.
5: In figure 3, the panel without FA shows the highest α value of 0.98 at 300 Hz, and the that with FA panel possess the lowest α value of ~0.45. The reinforcing filler fly ash seems the main influence factors of the sample’s acoustic absorption properties. Then, what is the effects of the oil palm fibers?
6: The literature number jump from 17 to 35 in the manuscript. There are also some typo errors in the text.
Reviewer 2 Report
The manuscript entitled "Development and characterization of natural fiber-based composite panels" presents an interesting experimental study conducted on the effect of fly ash addition in palm fibers composites with the resin matrix. However, the number of tested specimens wasn't presented and other issues must be addressed. The paper needs minor revisions before it is processed further, some comments follow:
Introduction section
The introduction section must be improved. A good introduction includes a comprehensive presentation of relevant previous studies, those studies must be quantitatively evaluated and presented, in order to show the current state of the art in the field. Please conduct and quantitative analyses of the previous publications as presented for reference 12 in lines 49-51.
Experimental work
Please provide the relevant characteristics of the raw materials. Please provide the particle size distribution, density and specific surface area of the fly ash. Also, please provide the purity of the NaOH flakes and other relevant information that is necessary to assure the experiment's repeatability.
Figure 1 – Please remove Figure 1 as it doesn’t have any scientific value. The equipment can be identified based on the description from line 183. Therefore, no necessary information/data can be extracted from this figure.
Results and discussions
Analysis of acoustic and damping properties of the panels
Please specify the number of tested samples. How many samples have been tested? The results were similar for all tested samples? Are the profiles overlapping?
Figure 8 and Figure 9 – Please introduce a scalebar on each figure.
General remark
The study presents interesting results that could conduct to the optimum use of fly ash in the industrial manufacture of reinforced panels. However, the homogeneity of the composites wasn’t addressed. Please introduce corresponding comments and evaluations related to the distribution of fly ash particles and fibers into the composites. Also, multiple specimens should be tested to obtain a clear overview of the behavior of these products.
Round 2
Reviewer 1 Report
The questions have been answered and the paper can be accepted.